# Assessing the Impact of Corporate Social Responsibility, Green Shared Vision on Voluntary Green Work Behavior: Mediating Role of Green Human Resource Management

**Jiang Yang** [1,*], **Saqib Yaqoob Malik** [2,3,*], **Yasir Hayat Mughal** [4] , **Tamoor Azam** [5], **Wajid Khan** [6] , **Muhammad Asif Chuadhry** [7], **Muhammad Ilyas** [2] **and Yukun Cao** [1]

1   College of Economics & Management, Northeast Forestry University, Harbin 150040, China; cyklk@163.com
2   Department of Management Sciences, Preston University, Islamabad 44000, Pakistan; ilyas.afridi1988@gmail.com
3   Department of Media Management, University of Religions and Denominations, Qom 37491-13357, Iran
4   Department of Health Administration, College of Public Health and Health Informatics, Qassim University, Al-Bukayriyah 52531, Saudi Arabia; y.hayat@qu.edu.sa
5   Department of Management Sciences and Engineering, School of Management and Economics, Kunming University of Sciences and Technology, Kunming 650000, China; tamoorazam@hotmail.com
6   Department of Business Management, University of Baltistan, Skardu 16200, Pakistan; wajid.khan@uobs.edu.pk
7   Department of Education and Leadership, Shifa Tameer-e-Millat University, Islamabad 44000, Pakistan; asif.epm@gmail.com
*   Correspondence: jiangyang@126.com (J.Y.); malikhashir58@yahoo.com (S.Y.M.)

**Abstract:** Due to increasing concerns and pressure from stakeholders, firms are eager to initiate green initiatives to produce ecofriendly products and services, which are less harmful for the environment. Consumers are willing to pay high prices for ecofriendly products and services. Thus, firms need a motivated workforce to achieve their green objectives. This is only possible if firms adopt their green policies in their green shared vision and start socially responsible activities to gain society's and stakeholders' attention, which is possible if firms start CSR activities at regular intervals. The purpose of this study was to examine the mediating role of "Green Human Resource Management" (GHRM) on the relation between corporate social responsibility (CSR), green shared vision (GSV), and voluntary green work behavior (VGWB). Employees of manufacturing companies were the participants of the present study and a non-probability convenience sampling technique was employed to determine the sample size. Data were gathered from manufacturing companies using a cross-sectional survey method. The total number of firms included was 100, and information on the firms included in the study included cement (10), sugar (26), leather (22), steel (8), food (21), beverages (2), furniture (3), construction (2), pharmaceutical (2), plastic (2), and dairy (2). The total number of questionnaires distributed among employees of the above-mentioned manufacturing firms was 700, and 500 completed questionnaires were used in the analysis, thus yielding a response rate of 71.42%. Using the smart PLS partial least square software (Version 4), the structural equation modeling (PLS-SEM) technique was applied for the statistical analysis. It was evident from the results that the measurement model had established convergent and discriminant validities. A structural model for testing hypotheses was established in the second step. Findings of the study revealed that CSR, GSV, and GHRM practices and VGWB were significantly related with each other. CSR, GSV, and GHRM have significant effects on VGWB. Additionally, it can be inferred from the results that GHRM significantly mediated the relationship between corporate social responsibility and green shared vision and VGWB. From Pakistan's perspective, the research study has applied and validated the natural resource-based view (NRBV) theory, and practitioners and researchers may benefit from its findings. This study has opened doors and paved a path for future studies to use this model and come up with interesting findings by adding more mediating variables. For any organization, it is imperative to have a motivated team which is capable enough to help firms to achieve their green targets. Hiring talented and hardworking employees and initiating socially responsible activities help firms to obtain a competitive advantage and enhance the VGWB of their employees.

**Keywords:** green shared vision; voluntary green work behavior; green human resource management; corporate social responsibility; natural resource-based view; signaling theory; stakeholder theory

## 1. Introduction

Interest in obtaining a competitive advantage and sustainable performance, as well as numerous other green concepts, has been increasing during the last few decades. These concepts are the result of human negligence in the manufacturing process, which has caused major harm to the environment. The existence of manufacturing firms with USD 1 billion in annual revenue confirms that the only way of survival in industry is environmental sustainability [1]. Furthermore, environmental protection has become an eminent part of the responsibilities of every other entity in commercial society [2]. Pressure from stakeholders (suppliers, creditors, consumers, employees, and government) to initiate green activities in firms' vision to reduce environmental issues is one of the main reasons [3]. Some of the prominent green practices, such as corporate social responsibility (CSR),green shared vision (GSV), and green human resource management practices (GHRM), are considered to be imperative for the progress of the companies and sustainability of the environment and for enhancing voluntary green work behavior (VGWB). Corporate social responsibility is the term used for the self-realization of manufacturing organizations in terms of adopting protective behavior against environmental harms caused by their intrinsic and extrinsic operational activities. It originates from the vision of higher management and is not taken as a costly tool but a strategic inventiveness, preferably adopted by manufacturing organizations to attain a high competitive advantage [4]. CSR has been in the limelight academically as well as industrially for more than half a century; however, its implementation has received less attention [5].

Green shared vision can be taken as a strategy that prompts all members to adopt green behavior in their own capacities. Ref. [6] proposed this concept and, in reference to [7], claimed GSV to be a prominent strategic direction of common sustainable targets and objectives that has been incorporated by employees of companies. Moreover, green shared vision provides the organization with purpose and a way of management so that a shared vision may prevail, in addition to an environment of organizational mindfulness and a process of mindful organizing for attaining sustainability within the company. CSR and GSV, being important to manufacturing organizations at present, require a voluntary motivation to be implemented by the employees, currently known as voluntary green work behavior. Both CSR and GSV have a significant impact on voluntary green work behavior (VGWB) in an organization. Within an organization, employees cannot be compelled to maintain ideal behavior (i.e., VGWB as a currently ideal behavior) with harsh and exhaustive rules and regulation, firstly because such rules will impact employee performance negatively [3]. Secondly, labor unions and improved labor rights demand better treatment of workers by these organizations [8]. At this point, green human resource management (GHRM) has come to be seen as an implicit steward for incorporating green practices in an organization. Additionally, a shared vision provides the organization with purpose and a way of management so that a shared vision may prevail, in addition to an environment of organizational mindfulness and a process of mindful organizing for attaining sustainability within the company [6]. Both GSR and GSV have a significant impact on voluntary green work behavior (VGWB) in an organization. Empirical evidence on the influence of CSR and GSV and the indirect effect of GHRM on individuals' behavioral outcomes is contradictory and inconclusive, and studies investigating the link between CSR, GSV, GHRM, and VGWB have numerous theoretical and methodological weaknesses. Increasing VGWB is the main issue for firms; firms use different strategies to motivate their workforce, which could help them to implement green activities, and for this purpose, a talented, hardworking workforce is required. This aim could be fulfilled by hiring those employees who have awareness and willingness to work voluntarily to reduce environmental issues. Secondly,

firms could increase their social responsibility activities, which could enhance their reputation in the eyes of stakeholders. Signaling theory explains this notion among two parties and states that the sender (the firm) must decide how and when to communicate or signal CSR, GHRM, and GSV, and on the other hand, the receiver (employees) must be able to decide how to interpret these signals. According to Garavan [3], the signaling framework states that signaling would influence the behaviors and attitude of employees, and the key behavior and issue in this study is VGWB.

According to [9], the literature currently suggests that employee attempts to innovate in a greener way may be motivated by green-oriented factors. Researchers in this discipline place emphasis on how management techniques or leadership ideologies (i.e., GHRM) may have a significant effect on how employees respond in relation to their surroundings [6]. GHRM operates as a role model to effectively engage employees' comparable green-oriented activities. GHRM has, therefore, been an eminent practice in the environmental management of industry. It refers to the comprehension of a nexus between the operations of an organization that impacts the natural environment and setting human resource management on the right path towards a sustainable environment [10]. GHRM plays an imperative role in the integration of several HR practices to ameliorate an organization's performance. Additionally, GHRM enhances efficiency and mitigates the risk of loss or high costs. Consequently, GHRM is significant for an organization in improving CSR, GSV, eco-friendly human resources, and gaining a good competitive edge over rivals. Further, ref. [11] claims that GHRM helps an organization achieve its ultimate goals by improving employees' behavior, conduct, attitude, and perceptiveness concerning the dignity and image of the organization. Thus, GHRM plays a vital role in the practices of management that deal with human resource activities in a commercial entity. Sustainability is presently being examined across the board in the context of GHRM. As per ref. [12], green HRM is the optimal exploitation of HRM policies to enhance the sustainable practice of resources within business enterprises to endorse the cause of environmentalism that will eventually lead to further amelioration of employee optimism and consummation. The rest of the literature holds GHRM to be the employment of "HRM policies, philosophies, and practices" to promote the sustainable use of business resources and thwart any untoward harm arising from environmental concerns in organizations [13].

The term "Green HRM" is currently all the rage in business and its significance is only going to increase over time [14]. This term has also emerged as a significant topic in recent research studies because of the growing worldwide awareness of environmental management and sustainable development within companies. The phrase "GHRM" today describes a concern for both the social and economic well-being of businesses as a whole, as well as for their employees. Even though there is a wealth of information on the topic of "Green HRM," there are still uncertainties on the best way to adopt green HR management practices in organizations throughout the globe in order to create a truly green corporate culture [3]. On the other hand, developing countries like Pakistan confront various challenges in incorporating sustainable regulation, owing to factors such as scarce capacity, insufficient resources, and lack of realization and awareness by officials [15].Therefore, there is a dire need for vigilant and proactive GHRM practice in Pakistan to cope with the sustainability issues within the manufacturing industry. The purpose of the current study was to investigate the mediating effect of green human resource management practices on the relationship between CSR, GSV, and voluntary green work behavior. The present study has been conducted to answer the following main research question. Can VGWB be enhanced by adopting GHRM practices, green shared vision, and corporate social responsibility?

The present study makes the following original contributions to the corpus of knowledge:

1.    Studies on green shared vision, CSR, and VGWB are limited.
2.    Green shared vision and CSR are used as predictors together for the first time.
3.    Empirical evidence from Pakistan's perspective regarding CSR, GSV, and GHRM needs to be reported.

The literature on GHRM in the scenario of Pakistan needs to be improved.

## 2. Literature Review

### 2.1. Voluntary Green Work Behavior (VGWB)

Environmental problems including pollution, environmental deterioration, and climate change present a serious challenge to humanity, giving organizations a new purpose, defending the natural environment, and working towards ecological sustainability [16,17]. Environmental concerns are gaining momentum among manufacturing firms worldwide [18]. By adopting and putting into effect green policies and practices, organizations are addressing their green initiatives. Green organizational practices are something that management and organizational researchers refer to as initiatives that enhance environmental sustainability, while green behaviors are actions that support environmental sustainability [19]. Green behavior is the degree to which employees carry out their duties in a way that protects the environment and saves resources, such as by adopting green practices or choosing eco-friendly alternatives [20]. However, a lot of green programs rely on individuals' voluntary involvement and participation. VGWB may be defined as employees' own acts that support the employer firm's commitment to environmental sustainability but are not governed by any formal environmental management systems or rules [18]. Further, according to [20], this includes employees' environmental actions that are voluntary and outside the scope of their jobs, such as turning off the electricity when leaving the workplace or recycling materials at work. VGWB may also be promoted by CSR [21], GHRM [22], and leadership [23].

Businesses are more environmentally conscious. Businesses worldwide are promoting environmental responsibility due to worries about environmental deterioration and climate change. Green policies can promote revenue, brand awareness, and employee outcomes [22]. Employees are responsible for implementing a company's green policy; therefore, companies must promote and change their behavior. "Green" employee conduct is environmentally sustainable [24]. Further, the authors of [25] believe that green acts must be considered for employment needs. And voluntary green workplace participation is a form of organizational citizenship [26]. Our theoretical framework is based on the literature on corporate social responsibility, which suggests that normative elements, such as congruence of values with issues, may impact social responsibility. Voluntary green behaviors have been studied outside the workplace for decades, but organizational experts have recently begun to explore the reasons [18]. Environmental psychologists have historically examined green behavior and its predecessors in non-work contexts, but it is important to study it in the workplace, since differences between work and non-work settings may limit generalizability [27].

Previous research shows that certain human and environmental elements may have an impact on an employee's green work behavior [20]. According to studies, job-related factors have a significant impact on the green behavior of employees. For instance, perceived organizational support [28], job satisfaction [29–31], and organizational commitment [32,33] have been found to have a positive influence on employees green behavior. Firms can encourage green behavior in their workforce by adopting and implementing certain policies and management strategies. A key predictor of employee green behavior is the firm's sustainability policy [34,35]. According to research, green human resource practices foster a green workplace culture and hence promote green behavior [24].

### 2.2. Theoretical Basis of the Study

The resource-based view (RBV) theory was first presented in 1991, although there has been criticism because it ignores the environment. The natural resource-based view theory (NRBV), which was established by [36], addresses environmental problems brought on by human irresponsibility. It handles environmental problems brought on by human carelessness. For a sustained competitive advantage, theresource-based theory highlights that a resource must be valued, scarce, and inimitable [37]. The NRBV claims that sustain-

able development, product stewardship, and pollution prevention are the three primary strategic competences. Each of them is influenced by various environmental factors, relies on multiple key resources, and derives its competitive advantage from various sources. Hence, it should be no surprise that over the past 15 years, the majority of NRBV applicability has been centered on preventing pollution [38]. In fact, one of the topics that is frequently discussed in studies on businesses and the environment is if and when it makes sense to go green [39]. The present study addresses the literature gap by examining the mediating role of GHRM between CSR, GSV, and voluntary green work behavior. This study contributes towards the NRBV theory. It is also essential to note this study's contribution to stakeholder theory. This theory is based on the principle that business organizations operate in an ecosystem of several stakeholders, and each stakeholder contributes towards the sustainability of the business and adds value to stakeholder groups. Likewise, it is evident that sustainable shareholder value cannot be attained without having a strong relationship with stakeholders, such as employees, customers, consumers, suppliers, and creditors. In addition, signaling theory explained the notion of the relationship between firms (sender) and receivers (employee); organizations have to send and communicate signals to their employees regarding VGWB and employees must be able to interpret those signals. Organizations use GHRM, CSR, and GSV as signals and raise the importance and significance of these constructs to enhance VGWB.

### 2.3. Corporate Social Responsibility

Corporate social responsibility (CSR) refers to organizational acts and policies that are particular to the context and take into account the hopes of stakeholders and the triple bottom line of environmental, economic, and social performance [40]. The concept of corporate social responsibility is the result of businesses' dedication to searching out ways to positively impact society or anyone who may be impacted by their social actions [41]. The CSR plan outlines a company's social and environmental responsibilities. And it covers both a company's internal activities and its social impacts. There is no agreement on the definition of CSR in the literature that has been studied [42]. CSR may be described as undetermined, geographical, dimensionally indefinite, social, and cultural [43]. CSR can mean several things depending on the situation, the person understanding it, or even the company working on it [42]. However, there may be some agreement on the acceptance of the social, economic, and environmental aspects of CSR, and it is difficult to develop a universal and unbiased CSR vision due to various contextual factors [44]. CSR refers to a commitment to enhancing network well-being through independent business decisions and the contribution of company resources. Analyzing CSR shows how organizations interact with their clients, suppliers, retailers, and other stakeholders. According to Carroll [45], CSR comprises the legal, financial, ethical, and philanthropic expectations the society has of organizations at a particular time [45]. Studies addressing how CRS influences HR practices are included [46,47]. Firms cannot attain their objectives just through regulations and control mechanisms. Hence, employee acceptance and support of such objectives is necessary for organizations [48]. Without the active involvement of human resources, CSR initiatives cannot be recognized by any company. According to CSR policies, all functional departments should pursue green initiatives [49]. CSR could be used to accomplish sustainable HRM. Researchers appear to be interested in a particular area right now, specifically the connection between CSR and green HRM [50]. CSR regulations are the main factor of GHRM initiatives in many firms as a result [51]. GHRM is"a subset of sustainable HRM, where the latter also includes CSR issues" [52]. By implementing GHRM procedures, the company makes it clearly known to its workforce that it is dedicated to the environmentally responsible and social green cause above and beyond any financial gain. Green recruitment and selection involves finding and choosing candidates who are aware of environmental issues, hence through tests making sure that workers have a favorable attitude toward environmental concerns [53]. Green training programs are intended to improve workers' familiarity with, expertise in, and skills in green activities, as well as the environment that

encourages all staff members to participate in green projects. And green training should put an emphasis on modifying attitudes and perceptions of commitment to green goals [54]. Green climates can be aided by an approach that not only integrates green comprehensive programs but links to performance management systems [53]. Numerous studies have looked into employee perceptions of companies' CSR programs, which determine how employees behave in the workplace and lead to productive behaviors [55]. Further, ref. [56] comes to the conclusion that employees display environmentally friendly behavior when they see their firm taking part in ecologically friendly activities. Thus, it can be said that a productive workplace environment has an impact on voluntary green work behavior and in turn positively affects green human resource management. It is the moral and social responsibility of firms to take care of society. This shows the ethical behavior of the firms. Social responsibilities help the firms to attract more investors and obtain a competitive advantage. Through CSR, firms can also attract hardworking and talented workers, which help the firms to achieve green objectives [56]. These concepts are in line with the findings of ref. [57], who reported that CSR was positively associated with employee green work behavior and also that CSR was positively correlated with employees' well-being. The findings from past studies reported the positive effect of CSR on the green behavior of employees [58]. In addition, there is positive effect of CSR on GHRM [59,60]. In the same way, the results of Ubeda-Garcia et al. [61] showed that CSR positively effects GHRM. On the basis of the above discussion, the following hypotheses are developed:

**Hypothesis 1a (H1a).** *CSR has a significant effect on voluntary green work behavior.*

**Hypothesis 1b (H1b).** *CSR has a positive effect on GHRM.*

*2.4. Green Shared Vision*

A shared vision can describe an organization's common standards and goals that guide its employees towards the future of the company [7]. Green shared vision is an approach that encourages each member to undertake environmentally friendly practices in their individual capacity. A collective vision gives a shared strategic approach that might expose conflicting goals. Encouraging green policies and initiatives now relies heavily on a shared vision of green management. GSV can be defined as a clear and shared strategic direction of combined environmental objectives which has been fully embraced by employees in an organization [7]. In the environmental era, an overall organizational concern with regard to corporate sustainability is more important for promoting green concepts within a company [62]. A shared vision and supportive organizational changes can be encouraged to stimulate green innovativeness. GSV is key component of attaining a competitive advantage [63]. Also, GSV is regarded as a prerequisite condition for a business's pro-environmental conduct [64]. Employees who share a green vision are more ready to embrace sustainable behaviors. Moreover, it can encourage workers to meet future sustainable development objectives, promoting ecological initiatives and inspiring people to achieve excellent performance and work while pursuing green behavior to create sustainable business practices. GSV inspires workers to foster proactive involvement in the creative process in order to foster intrinsic motivation and to create green products [63]. Employee actions employing a specific approach might be guided by a shared vision, which offers a collective strategic direction. For improving staff members' pro-environmental behavior, a shared vision of green management is essential [64]. Studies have shown that when a suitable GSV is created, green product development performance [65], green creativity [6], environmental performance of employees [66], and green product psychological ownership [67] are improved. An increasing number of businesses are starting to use green practices and strategies to improve the green work behavior of employees [68]. Also, findings have shown a positive association between green commitment and GHRM, which was increased by a high level of shared green knowledge [69]. Moreover, research has shown that GSV has a positive effect on green product performance [70], and GSVis also

positively associated with green self-efficacy, green creativity [6], and green innovation [71]. On the basis of the above discussion, the following hypotheses are developed:

**Hypothesis 2a (H2a).** *Green shared vision (GSV) has a positive effect on VGWB.*

**Hypothesis 2b (H2b).** *Green shared vision (GSV) has a positive effect on GHRM.*

### 2.5. Green Human Resource Management

GHRM ensures that companies effectually adopt more ecologically sound practices. GHRM can raise employees' environmental consciousness. When a firm's environmental concerns and its human resource goals are aligned, this is known as GHRM [72,73]. There is a greater awareness that the environmental effects of HRM procedures must be taken into account at every stage of the procedure, since HRM practices help organizations develop and maintain an environmental management system (EMS), which helps them achieve higher environmental performance [74]. GHRM is crucial for business management for a number of reasons, such as the advantages to the environment, retaining employees, and improving a firm's appeal. In the past, HRM literature concentrated on the impact of individual practices rather than the impact of a cluster of HRM practices on company performance [75]. The primary goal of the research study of [76] was to investigate the several facets of green human resource management in small and medium-sized enterprises in Pakistan. And results revealed that various characteristics (age, experience, and gender) influence green HR practices. Emphasizing the organization's commitment to sustainability during recruitment and taking into account candidates' environmental values during the hiring process are likely to improve [53]. Two categories of employee green behavior exist: task-related green behavior and voluntary green work behavior [20]. Task-related green behavior includes actions that are formally designated and defined as parts of a job [77]. And ref. [20] defined voluntary employee green work a behavior as green behavior that includes personal initiative going beyond company expectations.

Since green knowledge and green behaviors are recognized and rewarded in the workplace, it is reasonable to assume that GHRM practices will have a direct impact on how employees engage in them. However, GHRM practices might or might not have a direct impact on voluntary green work behavior., due to the fact that these behaviors are not formally acknowledged; rather, they are impacted by people's awareness of a firm's green culture and desire to participate in such behaviors [24]. The findings of previous research revealed that GHRM was positively related with task-related green work behaviors [22], and also, GHRM positively affects voluntary green behaviors [22]. On the basis of the above discussion, the following hypothesis is postulated:

**Hypothesis 3 (H3).** *GHRM has a positive effect on voluntary green work behavior.*

### 2.6. The Mediating Role of GHRM

Green HRM practices can be thought of as a collection of methodologies, initiatives, and policies that encourage employees in a business to adopt eco-friendly practices to build environmental sustainability and a socially conscious and resource-efficient work environment. Green HRM may be described as HRM practices and guidelines that support a business and, more importantly, seek to minimize harm caused by companies' adverse environmental actions [74]. GHRM refers to when the environmental priorities of the company and the objectives of human resources match [72]. It is unsafe for both the present and future generations because of carelessness and the excessive exploitation of natural resources [78]. Green HRM focuses on educating staff members about green practices and raising their level of environmental knowledge, efficiency, and performance. Green HRM is used as a mediator in this study because it is still a relatively new approach that includes activities like hiring and selecting employees, rewarding them, motivating

them, providing them with training and development opportunities, and conducting evaluations to foster an ecofriendly workforce. A previous research study also used GHRM as a mediator [61]. To our knowledge, no research study has yet taken GHRM as a mediator between corporate social responsibility and VGWB. And also, we have found no studies in which GHRM was used as a mediator between green shared vision and VGWB. Thus, the current study documented empirical findings of effects from GHRM between corporate social responsibility and VGWB and also from GHRM between green shared vision and VGWB. Therefore, on the basis of the above discussion, the following hypotheses are developed:

**Hypothesis 4a (H4a).** *CSR and VGWB are mediated by GHRM.*

**Hypothesis 4b (H4b).** *GSV and VGWB are mediated by GHRM.*

### 3. Materials and Methods

*3.1. Population, Sampling, and Data Collection Methods*

The goal of this research was to investigate the relationship between CSR, GSV, and voluntary green work behavior, in addition to the role of GHRM as a mediator. A review of the literature served as the foundation for creating the research model (Figure 1).

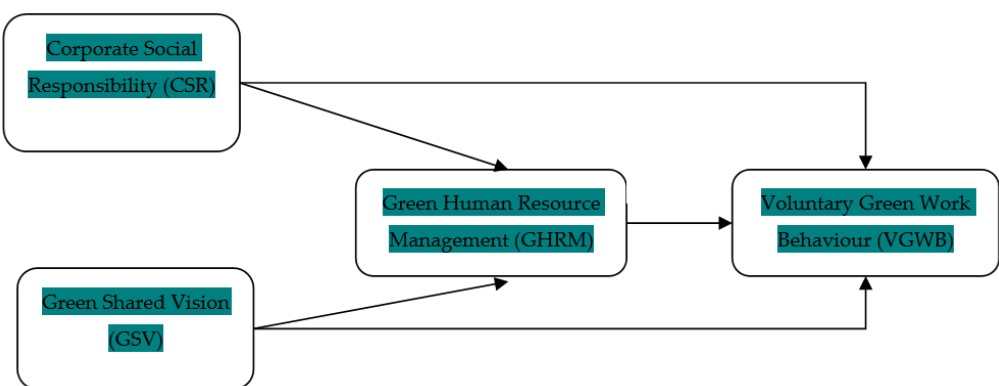

**Figure 1.** Research model and hypotheses.

*3.2. Inclusion Criteria of the Firms*

Researchers collected the data from the Pakistani manufacturing sector in 2022. The reasons for selecting Pakistani manufacturing firms are as follows. Limited empirical evidence is available on VGWB, GHRM, GSV, and CSR in Pakistani manufacturing firms. Pakistan's manufacturing sector does not have awareness about green initiatives and faces many challenges as compared to the manufacturing sectors of developed countries. Therefore, studying GHRM, GSV, VGWB, and CSR would extend our understanding about past studies' research findings. Firms were selected randomly (n=100). Inclusion criteria consisted of the following. Firms must be listed on the Pakistan stock exchange; the number of employees should be more than 100; the firm must have initiated green activities. Those firms which had initiated green activities had mentioned this on their websites, and those firms having received awards for sustainable performance or green initiatives were selected. The total number of firms included was 100; information of the firms included in the study include cement (10), sugar (26), leather (22), steel (8), food (21), beverages (2), furniture (3), construction (2), pharmaceutical (2), plastic (2), and dairy (2).

*3.3. Data Collection Methods*

The unit of analysis in this current study was individuals. A total of 100 firms were selected, and after obtaining permission from the director human resources or manager of human resources, the questionnaires were distributed to employees/respondents who

were involved directly in GHRM practices. The aim of the study and data collection was explained to respondents prior to data collection and the distribution of the questionnaire. Respondents were assured of the confidentiality of their data [79]. The questionnaire was distributed face to face. In order to avoid and reduce the risk of common method variance (CMV), we used 6 weeks' time lag. Data were collected in a two-stage/wave design. In the first wave, respondents were asked to provide data about demographic variables, i.e., personal details and CSR and GSV, and in the second, cohort data about GHRM and VGWB were collected. A total of500 completed questionnaires were received and used in the statistical analysis. A total of700 questionnaires were distributed among the employees and 500 completed questionnaires were used in the analysis, thus yielding a response rate of 71.42%.

### 3.4. Measures

The questionnaire distributed was in the English language. The team of researchers extensively discussed the applicability of constructs and measures from the perspective of Pakistan. Seven-point Likert scales were used for all measures. For CSR, a four-item scale was used, and for GSV, a four-item scale was adopted following Chang et al., 2020 [66]. For GHRM, ashort-form questionnaire was adopted with 6 items [24,80], and for VGWB, a 3-pointscale was adopted, following ref. [3].

### 3.5. Data Analysis Techniques

Smart PLS-SEM 3 was used in the analysis. This is 2nd-generation software used for non-normal data and small datasets. It is variance-based software. It is used for confirmatory factor analysis to check the reliability and validity of the scales, i.e., convergent and discriminant validities. Secondly, it is used to test hypotheses in structural models. Bootstrapping, BCILL and BCIUL, t-statistics, and significance values help the researcher to test hypotheses.

Regarding age, the majority of the participants (350, 70%) were 25–34 years old, followed by the age group of 16–24 years old (111, 22.2%). Only 38 participants were aged between 35 and 44 years, and one was more than 44 years old. Most of the participants were male, at 339 (67.8%), followed by females at 161 (32.2%). Regarding education, most of the respondents had a master's degree (436, 87.2%), followed by those having a bachelor's degree (62, 12.4%). Only two respondents had PhD degrees. Participants were also asked to mention their employment type; 311 (62.2%) were working in public sector organizations and 189 (37.85) in the private sector (see Table 1).

**Table 1.** Demographic information.

| Variable | Characteristics | n | Percentage % |
|---|---|---|---|
| Age | 16–24 | 111 | 22.2 |
| | 25–34 | 350 | 70.0 |
| | 35–44 | 38 | 7.6 |
| | 44+ | 1 | 0.2 |
| Gender | Male | 339 | 67.8 |
| | Female | 161 | 32.2 |
| Education | Bachelor | 62 | 12.4 |
| | Master | 436 | 87.2 |
| | PhD | 2 | 0.4 |
| Employment | Public | 311 | 62.2 |
| | Private | 189 | 37.8 |

### 3.6. Analytical Strategy

The analysis of the data was conducted with PLS-SEM. Researchers prefer to use PLS-SEM for its advanced tools for analysis of big data as well as small datasets. Prior to testing the hypotheses, we conducted confirmatory factor analysis (CFA) to investigate whether the measurement model was fit or not. The findings in Table 2 indicate that composite reliability met the threshold of 0.7 (Hair et al., 2017) [81]. Cronbach's Alpha met the cut-off level of >0.70 (Field, 2013) [82], and AVE met the standard criterion of >0.50. Factor loadings > 0.70 (Table 2 and Figure 2). We have also provided co-linearity results, i.e., VIF, with all values of all items below standard criteria, i.e., <5. Discriminant validity is presented in Table 3; thus, reliability, validity, and discriminant validity are not an issue in this study.

**Table 2.** Measurement model confirmatory factor analysis.

| Items | Loadings | CR | AVE | $\alpha$ | VIF |
|---|---|---|---|---|---|
| CSR1 | 0.805 | 0.907 | 0.710 | 0.864 | 1.959 |
| CSR2 | 0.778 | | | | 1.828 |
| CSR3 | 0.883 | | | | 2.960 |
| CSR4 | 0.898 | | | | 3.148 |
| GSV1 | 0.882 | 0.898 | 0.745 | 0.829 | 2.095 |
| GSV2 | 0.861 | | | | 1.833 |
| GSV3 | 0.846 | | | | 1.838 |
| GHRM1 | 0.760 | 0.914 | 0.639 | 0.886 | 1.890 |
| GHRM2 | 0.861 | | | | 3.811 |
| GHRM3 | 0.758 | | | | 1.972 |
| GHRM4 | 0.817 | | | | 2.446 |
| GHRM5 | 0.766 | | | | 2.165 |
| GHRM6 | 0.827 | | | | 3.153 |
| VGWB1 | 0.851 | 0.872 | 0.694 | 0.786 | 1.609 |
| VGWB2 | 0.772 | | | | 1.639 |
| VGWB3 | 0.872 | | | | 1.684 |

**Table 3.** HTMT ratios' discriminant validity.

| Variables | 1 | 2 | 3 |
|---|---|---|---|
| CSR | | | |
| GHRM | 0.752 | | |
| GSV | 0.712 | 0.984 | |
| VGWB | 0.066 | 0.393 | 0.465 |

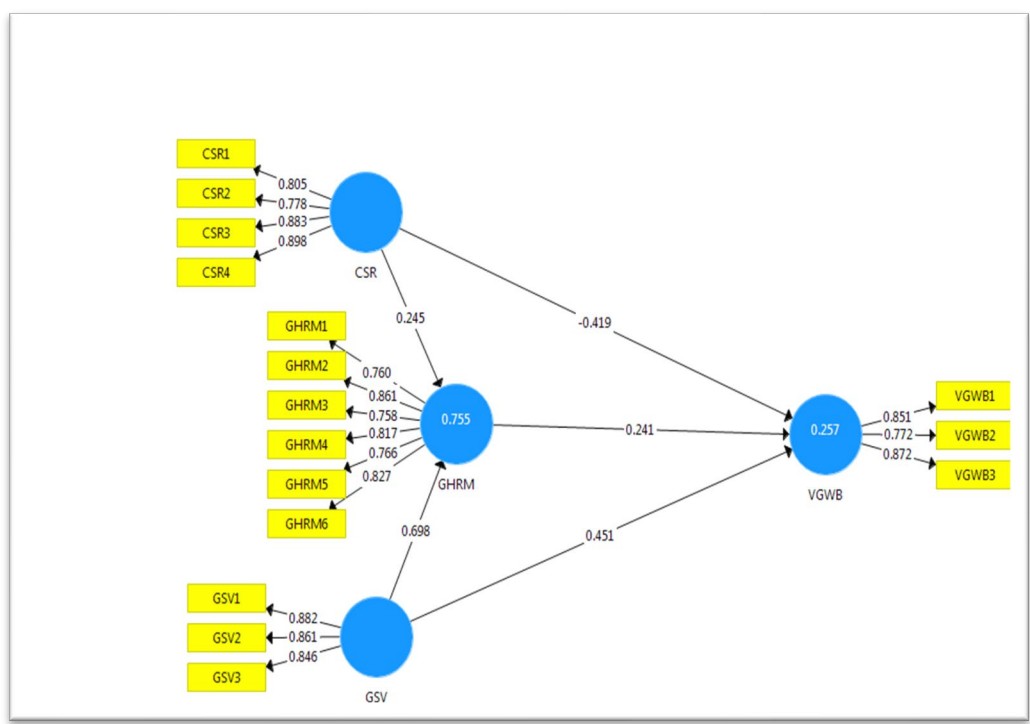

**Figure 2.** Measurement model.

## 4. Results

Table 4 presents the findings on the direct effects. It is evident that there is significant relationship between CSR and GHRM ($\beta = 0.245$ ***, $p < 0.01$); CSR and VGWB ($\beta = -0.360$ ***, $p < 0.01$); GHRM and VGWB ($\beta = 0.241$ ***, $p < 0.05$); GSV and GHRM ($\beta = 0.698$ ***, $p < 0.01$); GSV and VGWB ($\beta = 0.619$ ***, $p < 0.01$).We thus found support for Hypotheses (1)–(4). A negative sign shows the direction of the relationship. In Table 4, CSR has a negative impact on VGWB. It means that an increase in one variable would possibly decrease the other variable (Field (2013)) [82].

**Table 4.** Direct effect hypothesis testing.

| Hypotheses | β | SE | T | $p$ | BCILL | BCIUL | Support |
|---|---|---|---|---|---|---|---|
| CSR → GHRM | 0.245 | 0.036 | 6.890 | 0.000 | 0.169 | 0.304 | Yes |
| CSR → VGWB | −0.360 | 0.049 | 7.343 | 0.000 | −0.460 | −0.273 | Yes |
| GHRM → VGWB | 0.241 | 0.083 | 2.897 | 0.004 | 0.081 | 0.411 | Yes |
| GSV → GHRM | 0.698 | 0.029 | 24.405 | 0.000 | 0.641 | 0.755 | Yes |
| GSV → VGWB | 0.619 | 0.046 | 13.426 | 0.000 | 0.519 | 0.703 | Yes |

Table 5 presents the findings on indirect effects, i.e., mediation analysis. There is a significant indirect effect of GHRM on CSR and VGWB ($\beta = 0.059$ **, $p < 0.05$), and a significant mediating effect of GHRM on GSV and VGWB ($\beta = 0.168$ **, $p < 0.05$). Thus, we found support for our mediation analysis. According to Hair et al. (2017) [82], if all direct and indirect path coefficients are significant, it is called complementary mediation (see Figure 3).

**Table 5.** Indirect effects.

| Indirect Hypotheses | β | SE | T | *p* | BCILL | BCIUL | Support |
|---|---|---|---|---|---|---|---|
| CSR → GHRM → VGWB | 0.059 | 0.023 | 2.586 | 0.010 | 0.019 | 0.110 | Yes |
| GSV → GHRM → VGWB | 0.168 | 0.058 | 2.882 | 0.004 | 0.059 | 0.283 | Yes |

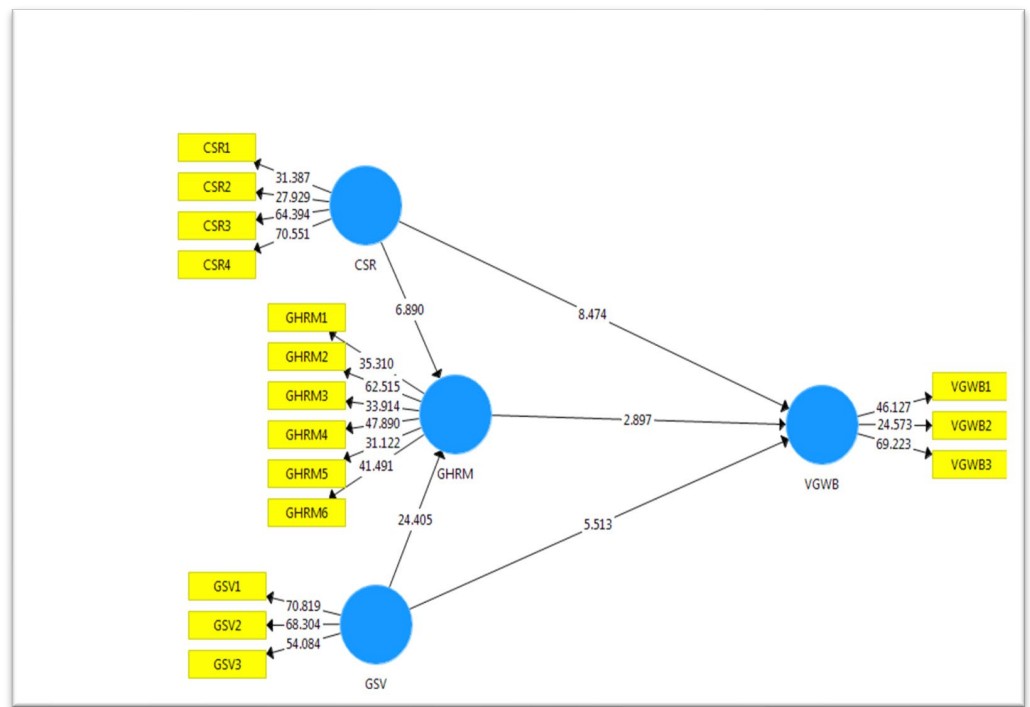

**Figure 3.** Structural model.

## 5. Discussion

H1a was developed to observe the effect of CSR on VGWB, whereas H1b was established to determine the effect of CSR on GHRM. The results revealed that H1a and H1b were significant. Corporate social responsibility has a significant effect on VGBW and GHRM. The findings of the current study support H1a, showing how CSR can significantly improve green workplace behavior in companies. CSR positively affects VGBW. These findings are consistent with the previous findings of ref. [57], who used social exchange theory and PLS-SEM for data analysis in their study and reported that CSR was positively related to employee green behavior, and also that CSR was positively related to employee well-being. Furthermore, the findings also show that CSR positively influences the green behavior of employees [58]. CSR has become embedded in daily operations; businesses are now reaping both direct and indirect benefits from its positive effects on employee well-being. A healthy workplace atmosphere improves employee well-being, which in turn helps them to develop and control their green work behavior.

The results of the recent study also support H1b. This shows that CSR may greatly enhance green behavior and significantly affect GHRM. According to the instrumental view of CSR-HRM, the interaction between CSR and GHRM is intended to improve organizational performance [17]. CSR has a significant role in motivating employee green behavior [21]. So, firms should encourage inventive and green behavior of the workforce. Workers should be urged to support sustainability projects and to adopt green behavior and an ecologically friendly work ethic as well. The conclusions of earlier studies were inline with the idea that CSR and the green behavior of employees are positively related [55,83]. The results of this research supported H1b. The findings of the existing study show that green CSR has a positive effect on GHRM and supported H1b. These findings were consistent with



the previous findings of [59,60], who revealed that CSR has a positive effect on GHRM. Likewise, the results of [61] revealed that CSR positively influences GHRM. All functional departments must implement green initiatives, according to CSR regulations. And CSR can be used to attain sustainable HRM [17]. The relationship between CSR and GHRM is vibrant and participatory. Generally, CSR positively impacts GHRM. Thus, managers should grasp opportunities to support CSR and GHRM initiatives inside the firm.

H2a was established to examine the impact of green shared vision on VGBW, whereas H2b was developed to determine the effect of green shared vision on GHRM. Green shared vision has a positive impact on VGBW and GHRM. The findings showed that H2a and H2b were significant. Green shared vision (GSV) is an approach that encourages each member to undertake environmentally friendly practices in their individual capacity. The results of the current study supported H2a. The findings are similar to previous findings that GSV positively influences green product performance [70], and also that GSV is positively associated with green creativity and green self-efficacy [6]. Green shared vision can foster organizational awareness, since it gives the organization a sense of direction and goal. Furthermore, GSV can be utilized to motivate team members to increase their readiness to go above and beyond expectations. According to studies, when suitable GSV is established, green commitment [84], sustainable performance [85], and green innovation [71] are increased.

The results of the current study also support H2b, showing that GSV may significantly affect GHRM. GSV encourages employees to engage in green practices, as GSV is crucial for gaining a competitive advantage. Greener businesses motivate their employees to share a common green vision [6]. The findings of this study show that GSV has a positive effect on GHRM, which is consistent with previous studies that indicated that GSV positively influences green product psychological ownership [67], green organizational identity [67], and green creativity [84]. Businesses must broaden their shared vision for better GHRM and sustainability in firms. Also, results indicated a positive association between green commitment and GHRM, which is increased by a high level of green shared knowledge [69].

H3 was developed to observe the effect of green HRM on VGWB. GHRM has a significant effect on VGWB. The findings of the current study support H3, showing how GHRM can significantly develop VGWB in firms. Green management places a strong emphasis on communicating to employees about environmental objectives and developing competitive advantages depending on those environmental concerns. In order to encourage workers to carry out their work in the most environmentally responsible manner possible, green HRM is based on an ecologically friendly approach and attempts to develop voluntary green work behavior and green culture. GHRM positively affects VGBW. These findings are consistent with the previous findings of ref. [22], which showed that green HRM positively impacts voluntary green behaviors. Also, GHRM was positively associated with task-related green behaviors [22]. Green behaviors are recognized and appreciated in the workplace, so it is reasonable to assume that GHRM policies will have a direct impact on voluntary green behaviors.

Furthermore, Hypothesis H4a was generated to examine whether GHRM mediates the relationship between CSR and VGWB. Furthermore, Hypothesis H4b was developed to observe whether GHRM mediates the relationship between GSV and VGWB. The results show that GHRM significantly mediated the relationship between CSR and VGWB, thus supporting H4a. Moreover, the results also revealed that GHRM significantly mediated the relationship between GSV and VGWB. The findings of the current study also support H4b. As a research study, this work provides a novel contribution to the body of available literature on GHRM as a mediator of the relationship between CSR and VGWB. Based on the findings of the current study, Hypotheses H4a and H4b are accepted.

## 6. Conclusions

This study investigated the relationship between CSR, GHRM, GSV, and VGWB. In addition, this study also investigated the mediating effect of GHRM on the relationship

between CSR and VGWB and GSV and VGWB. GHRM mediates the relationship between CSR, GCV, and VGWB. VGWB is positively predicted by GHRM and GSV but negatively predicted by CSR. Thus, the researchers derived two main conclusions. Implementing GHRM practices gives an indication to employees that VGWB is appropriate and expected. One greener objective in their green shared vision enhanced employees' VGWB. Therefore, GHRM and GSV are found to be more dominant in enhancing employees' GVWB, followed by CSR and GHRM. This research study has applied a unique and different theoretical approach to understanding employees VGWB. This study has advanced the literature of CSR, GSV, GHRM, and VGWB. The researchers found support for direct and indirect hypotheses. The researchers have advanced GHRM as a novel mediator to enhance the VGWB of employees. The researchers have argued that there is scope to make use of a unique theoretical approach to better understand the subject matter.

### 6.1. Theoretical Implications

The contribution and originality of the current study lie in investigating the relationship between CSR, green shared vision, green HRM, and voluntary green work behavior (VGWB). Limited evidence has been reported, specifically from Pakistan's perspective, regarding these concepts. Through the lens of NRBV and stakeholder theory, this study hypothesized that GHRM would have a positive and significant indirect effect on the relationship between CSR, GSV, and VGWB. The findings indicated that the direct and indirect effects are significant. This implies that through CSR, GSV, and GHRM, voluntary green work behvaior could be enhanced. This study has contributed and extended the body of knowledge on CSR, GSV, GHRM, and VGWB. Moreover, the existing study has validated the scales of CSR, GSV, GHRM, and VGWB from Pakistan's perspective. In addition, this study has applied and validated NRBV and stakeholder theories in a developing economy, and researchers and practitioners can benefit from the findings of this study.

### 6.2. Practical Implications

Due to limited human and financial resources, an increase in environmental issues, and pressure from stakeholders, firms are more interested in offering ecofriendly products and services; hence, through implementing GHRM and GSV and conducting CSR activities, managers and policymakers of firms can enhance the VGWB of their employees. Employees would also align their objectives with firms' objectives. The manufacturingsector is a major contributor to theeconomy as well as a huge source of increasing pollution, but unfortunately, manufacturing firms have limited knowledge about the benefits of going green and CSR; hence, it is essential for managers and policy makers to raise awareness about the importance of CSR, GSV, GHRM, and VGWB. Moreover, firms could gain knowledge about how to address energy waste, water use, and aenvironmental issues by implementing green activities.

### 6.3. Limitations and Future Research Directions

The data were collected from a single source in this study, which could be seen as a significant source of bias. It is therefore recommended to use a mixture of methods in future studies. Longitudinal data could also be used to obtain a better understanding of the problem. The researchers used one mediator in this study; therefore, it is recommended to use two mediators in future studies, such as green intellectual capital and information integration. The data used in this study belong to a specific sector, so one should be careful while generalizing the findings to other sectors in other countries. The model could be applied in other sectors, such as hospitality, tourism and leisure, or educational institutions to gain further insight. In future study, the authors would like to add socio-cognitive characteristics of employees, such as reflective moral attentiveness (RMA), through the lens of signaling theory. It is believed that employees high in RMA have high VGWB and vice versa. Moreover, future researchers could also investigate the current framework by

dividing this framework into two levels, individual and organizational levels, for multilevel data analysis. The authors recommend HLM software.

**Author Contributions:** S.Y.M. and Y.H.M. conceptualized the study, wrote the original draft and methodology, and performed the formal analysis, validation, and data curation. T.A. reviewed and edited the manuscript. W.K. and M.A.C. organized the database and reviewed and edited the manuscript. M.I., Y.C. and J.Y. wrote the literature sections. All authors have read and agreed to the published version of the manuscript.

**Funding:** The study presented in this paper was funded by the key research topics for economic and social development in Heilongjiang Province (22JYB231).

**Institutional Review Board Statement:** The ethical committee of (IRB & EC) the Faculty of Social Sciences and Humanities (FSSH) of ShifaTameer-e-Millat University, Islamabad has reviewed and approved this study. The study was conducted in accordance with the declaration of Helsinki.

**Informed Consent Statement:** Informed consent was obtained from all subjects involved in the study.

**Data Availability Statement:** The data supporting the conclusions of this article will be made available by the authors.

**Acknowledgments:** We would like to thank all the participants of the study.

**Conflicts of Interest:** The authors declare that they have no conflict of interest.

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
