# Peer review of "Assessing the Impact of Corporate Social Responsibility, Green Shared Vision on Voluntary Green Work Behavior: Mediating Role of Green Human Resource Management"

_sustainability, doi:10.3390/su152316398_

Round 1
Reviewer 1 Report
Comments and Suggestions for Authors
Dear Authors,
The paper topic is very interesting and relevant in contemporary business environment. However, you could improve theoretical background, give more theoretical and practical implications as well as some recommendations for future own and other researches. You should improve the logical coherence, strength of argument and academic soundness, as well.
Best wishes
Author Response
Reviewer 1
The paper topic is very interesting and relevant in contemporary business environment. However, you could improve theoretical background, give more theoretical and practical implications as well as some recommendations for future own and other researches. You should improve the logical coherence, strength of argument and academic soundness, as well.
Response: Authors want to thank reviewer for valuable comments and suggestions. As per suggestions theoretical background is improved, more theoretical and practical implications are added in section 6.1 and 6.2, recommendations for future own and other researchers are also written and highlighted in yellow. Logical coherence and more theoretical underpinning are also given in heading 2.2. Following are the new information which is added in the manuscript:
Heading 2.2 Theoretical basis of the study
Another contribution is also essential to be reported as this study also contributed in stakeholder theory. This theory is based on the principle that business organizations operates in an ecosystem of several stakeholders, and each stakeholder contribute towards sustainability of the business and add value to stakeholder group. Likewise it is evident that sustainable shareholder value cannot be attained without having strong relationship with stakeholders such as employees, customers, consumers, suppliers and creditors. In addition, signaling theory explained the notion between firms (sender) and receiver (employee); organizations have to send and communicate signals to their employees regarding VGWB and employees must be able to interpret those signals. Organizations use GHRM, CSR and GSV as signals and raise importance and significance of these constructs to enhance VGWB.
6.1 Theoretical Implications
Contribution and originality of the current study lie in investigating the relationship among CSR, green shared vision, green HRM and voluntary green work behavior (VGWB). limited evidences are reported specifically in Pakistan’s perspective regaridng these concepts. Through the lens of NRBV and stakeholder theory this study hypothesized that GHRM would have positive and significant indirect effect on the relationship between CSR, GSV and VGWB. The findings indicated that direct and indirect effects are significant. This implies that through CSR, GSV and GHRM voluntayr green work behvaior could be enhanced. This study has contributed and extended the body of knowledge of CSR, GSV, GHRM and VGWB. Moreover, the existing study has validated the scales of CSR, GSV, GHRM and VGWB in Pakistan’s perspective. In addition this study has applied and validated NRBV and stakeholder theories in an developing economy and researchers and practitioenrs can take benefits from the findings of this stud.
6.2Practical Implications
Due to limited human and financial resoruces and increase in environmental issues and prressure from stakeholders firms are more concerned to offer ecofriendly products and servcies ,hence through implmenting GHRM and GSV, and conducting CSR activities managers, policymakers of the firms can enhance VGWB of their employees. Employees would also allign their objectives with firms’ objecitves. Manufacturign sector is major contributor towards economy as well as huge soruce of increasing pollution but unfortunalety the manufacturing firms have limited knowledge about benefits of going green and CSR hence it is essential for managers, and policy makers to raise awareness about importace of CSR, GSV, GHRM and VGWB. Moreover, firms could get knowledge how to reduce enegry waste, water resoruces, and decresse environmental issues by implementing grene activities.
6.3 Limitations and Future Research Directions
In future study, the authors would like to add socio-cognitive characteristics of employees such as reflective moral attentiveness (RMA) through lens of signaling theory. It is believed that employees high in RMA have high VGWB and vice versa. Moreover future researchers could also investigate the current framework by dividing this framework into two levels, individuals and organizational levels, for multilevel data analysis, authors’ recommends HLM software.
Reviewer 2 Report
Comments and Suggestions for Authors
The authors conducted an interesting scientific study. In section 3.3. Data Collection Methods, reasonable selection of respondents. Smart PLS partial least square software, structural equation modelling (PLS-SEM) technique was applied for the statistical analysis. It was clear from the results that the measurement model had established convergent and discriminant validities. A structural model for testing hypotheses was established in the second step.
However, the article needs to be revised:
In the Abstract it is noted: "...It was established that VGWB was related to both CSR and GSV..." However, according to the purpose, should a connection be established with GHRM?
The purpose of the research should be clearly defined in the introduction. "An effort was made..." does not form the purpose of the research.
The purpose of the research is formed only on page 8 in section 3. Materials and Methods.
Citation requirements should also be cited: (3) (Huang et al., 2015); (6); (Hart, 1995)(36); (Carroll, 1991). There are also sentences that are not clear from the point of view of citation, for example, "...Environmental concerns are a top concern worldwide (18)..." What is the scientific significance of this citation?
2. Literature Review Contains 6 subsections, but it is not clear from the given information: whether there are quality scientific schools, which scientists have researched various aspects of this topic, these are similar studies. Literature Review is descriptive in nature with references to sources.
It is also surprising to see in the literature review "...And findings of also exposed that CSR positively influences green behavior of employees (58). Previous research findings also shown that CSR positively influences GHRM (59, 60). Also, the result (61) shown that CSR positively effects GHRM...." This is convenient as relevance and on the basis of which the purpose of one's own research can be formed.
It is also not entirely clear why the authors refer to the formation of the research hypothesis in section 2. Literature Review: "...
In light of this, the following hypotheses are made: Hypothesis 1a: CSR has a significant effect on voluntary green work behavior. Hypothesis 1b: CSR has a positive effect on GHRM...."
Comments on the Quality of English LanguageThe text needs editing. Articles should be added, for example, "of responsibilities", "especially in manufacturing" and etc.
Author Response
Reviewer 2
The authors conducted an interesting scientific study. In section 3.3. Data Collection Methods, reasonable selection of respondents. Smart PLS partial least square software, structural equation modelling (PLS-SEM) technique was applied for the statistical analysis. It was clear from the results that the measurement model had established convergent and discriminant validities. A structural model for testing hypotheses was established in the second step.
Response: Thanks and appreciate the comment of the respected reviewer.
However, the article needs to be revised:
In the Abstract it is noted: "...It was established that VGWB was related to both CSR and GSV..." However, according to the purpose, should a connection be established with GHRM?
Response: Thank you so much for the valuable comment. Yes hypotheses 1b, 2b and 3 were developed to investigate the relationship and effects with GHRM. These hypotheses are highlighted in red color. In addition these are also incorporated in the abstract and highlighted in red color.
The purpose of the research should be clearly defined in the introduction. "An effort was made..." does not form the purpose of the research.
The purpose of the research is formed only on page 8 in section 3. Materials and Methods.
Response: thank you so much of the valuable comments. The purpose of the research has been incorporated at page 4 at the end of the introduction section and highlighted in red color.
Citation requirements should also be cited: (3) (Huang et al., 2015); (6); (Hart, 1995)(36); (Carroll, 1991). There are also sentences that are not clear from the point of view of citation, for example, "...Environmental concerns are a top concern worldwide (18)..." What is the scientific significance of this citation?
Response: Thank you so much for correcting us. This would definitely increase the value of the manuscript. As suggestion authors have corrected these suggestions. At page 2 authors decided to keep the citations number 3 as it is more related with VGWB, at page 4 and 5 authors have corrected the numbering of citations for Hart 1995 and Carroll 1991 also the sentence has been corrected as follows at page 4
Environmental concerns are gaining momentum among manufacturing firms worldwide (18).
- Literature Review Contains 6 subsections, but it is not clear from the given information: whether there are quality scientific schools, which scientists have researched various aspects of this topic, these are similar studies. Literature Review is descriptive in nature with references to sources.
Response: Thanks for asking the explanation of the literature review section. Literature review is taken from past published studies which are published in well reputed web of science and Scopus indexed journals and cited by numerous authors. The citations represent the authenticity of the articles which are related with the current studies. Therefore authors have cited these studies to get support for the current study framework and hypotheses development.
It is also surprising to see in the literature review "...And findings of also exposed that CSR positively influences green behavior of employees (58). Previous research findings also shown that CSR positively influences GHRM (59, 60). Also, the result (61) shown that CSR positively effects GHRM...." This is convenient as relevance and on the basis of which the purpose of one's own research can be formed.
Response: As suggested by respected reviewer the above mentioned sentences at their respective place page number 6 are corrected in a more academic way so that readers can easily understand the meaning of these sentences. Moreover some explanation in light of past study has also been added before these sentences.
It is the moral and social responsibility of the firms to take care of societies. This shows the ethical behavior of the firms. Social responsibilities help the firms to attract more investors and obtain competitive advantage. Through CSR firms can also attract hardworking and talented workers which help the firms to achieve green objectives (56). These concepts are in line with findings of (57) reported that CSR positively associated to employee green work behaviour and also CSR positively correlated with employees wellbeing. The findings from past studies reported the positive effect of CSR on green behavior of employees (58). In addition, there is positive effect of CSR in GHRM (59, 60). In the same way, the result of Ubeda-Garcia et al., (61)
It is also not entirely clear why the authors refer to the formation of the research hypothesis in section 2. Literature Review: "...
In light of this, the following hypotheses are made: Hypothesis 1a: CSR has a significant effect on voluntary green work behavior. Hypothesis 1b: CSR has a positive effect on GHRM...."
Response: Thanks for raising this concern. Actually authors want to write the following sentence:
On the basis of above discussion, the following hypotheses are developed: this sentence has been corrected.
Comments on the Quality of English Language
The text needs editing. Articles should be added, for example, "of responsibilities", "especially in manufacturing" and etc.
Response: Thanks for comments. We have already corrected the English of the manuscript but as per suggestion of the respected reviewer the authors again carefully checked the text for spelling and grammar mistakes in the whole manuscript, wherever text needs correction authors have corrected and highlighted in pink color. If any changes required authors would be happy to correct the text again.
Reviewer 3 Report
Comments and Suggestions for Authors
Dear authors,
The manuscript is interesting. I would recommend highlighting the scientific problem in the introduction. After formulating the scientific problem, the conclusions should be adjusted accordingly. There are a few co-authors who, in my opinion, have quoted themselves too many times. Section 3.3 states that the questionnaires are given directly to respondents (face to face), although the return rate is later reported at 52 percent or 48 the second time. It seems strange. As is the number of respondents of exactly 500. Although random sampling was applied, at least some proportion of the selected organizations should have been provided. Analyzing the demographic data, it seems that only administrative workers were surveyed, as most of them have a master's degree. I recommend supplementing the methodology and reviewing the data, unifining list of references.
sincerely
Author Response
Reviewer 3
The manuscript is interesting. I would recommend highlighting the scientific problem in the introduction. After formulating the scientific problem, the conclusions should be adjusted accordingly. There are a few co-authors who, in my opinion, have quoted themselves too many times. Section 3.3 states that the questionnaires are given directly to respondents (face to face), although the return rate is later reported at 52 percent or 48 the second time. It seems strange. As is the number of respondents of exactly 500. Although random sampling was applied, at least some proportion of the selected organizations should have been provided. Analyzing the demographic data, it seems that only administrative workers were surveyed, as most of them have a master's degree. I recommend supplementing the methodology and reviewing the data, unifining list of references.
Response: Thank you very much for the comments respected reviewer. As suggested the problem of the study is highlighted in the introduction section in green color. Related articles are cited by co-authors to support the current study, its framework in order to have strong theoretical and logical coherence. In section 3.3 sentence have been corrected and response rate is also corrected which is 71.42%. Moreover, information about selected firms is added in green color in section 32. Those employees working in manufacturing sector and have awareness and knowledge of green initiatives have been contacted and participated in the survey. As, unit of the analysis is individual therefore, employees holding administrative and non-administrative positions can take part in the study. Following are the corrections added:
Total number of firms included were 100, information of the firms included in the study were cement (10), sugar (26), leather (22), steel (8), food (21), beverages (2), furniture (3), and construction (2), pharmaceutical(2), plastic(2), and dairy(2). Total number of questionnaire distributed among employees of above mentioned manufacturing firms are 700 and 500 completed questionnaires were used in the analysis thus yielding response rate of 71.42%
Empirical evidence on influence of CSR, GSV and indirect effect of GHRM on individuals’ behavioral outcome are contradictory and inconclusive, and studies investigating link between CSR, GSV, GHRM and VGWB have numerous theoretical and methodological weaknesses. Increasing VGWB is the main issue for firms, firms use different strategies to motivate their workforce which could help them to implement green activities and for this purpose talented, hardworking workforce is required and this aim could be fulfilled by hiring those employees who have awareness and willingness to work voluntary to reduce environmental issues, secondly, firms could increase their social responsibilities activities it could enhance their reputation in eyes of stakeholders. Signaling theory explains this notion among two parties and stated that sender (the firm) must decide how and when to communicate or signal CSR, GHRM and GSV and on the other hand received (employees) must be able to decide how to interpret these signals. According to Garavan (3) signaling framework states that signaling would influence behaviors and attitude of employees and they key behavior and issue in this study is VGWB.
Total number of firms included were 100, information of the firms included in the study were cement (10), sugar (26), leather (22), steel (8), food (21), beverages (2), furniture (3), and construction (2), pharmaceutical(2), plastic(2), and dairy(2).
Total 500 completed questionnaires were received and used in the statistical analysis. Total 700 questionnaires were distributed among the employees and 500 completed questionnaires were used in the analysis thus, yielding the response rate of 71.42%.
Reviewer 4 Report
Comments and Suggestions for Authors
Dear Authors,
First of all, it is a good paper but you have the recheck especially the abstract and the introduction's first paprgraph for minor English aditing as the reader can sometimes have difficulty while reading. (+punctuation).
Second point: Be clear in the parts 3.2 and 3.3. about the sample. I think its 500 individuals from 100 firms (clearly explain this, 5 from each or random).
The choice of PLS_SEM for this data set is a good choice. The discussion and conclusions are ok!
Best Regards,

Comments on the Quality of English LanguageI think the text might need minor English editing, especially the abstract and introduction's first part.
Author Response
Reviewer 4
First of all, it is a good paper but you have the recheck especially the abstract and the introduction's first paprgraph for minor English aditing as the reader can sometimes have difficulty while reading. (+punctuation).
Response: Thank you very for appreciation and according to suggestions first paragraph of abstract and introduction section are corrected and highlighted in purple color.
Abstract: Due to increasing concern and pressure from the stakeholders, firms are eager to initiate green initiatives to produce ecofriendly products and services, which are less harmful for environment. Consumers are willing to pay high prices for ecofriendly products and services. Thus, firms need motivated workforce to achieve its green objectives.
Introduction: Increase in interest to obtain competitive advantage and sustainable performance numerous green concepts have been evolved during last few decades. These concepts are results of human negligence in manufacturing process which caused a major harm to environment. Manufacturing firms with $1 billion annual revenue affirms that only way of survival in the industry is environmental sustainability. (1). Besides, environmental protection has become an eminent part of responsibilities of every other entity in the commercial society(2). Pressure from stakeholders (suppliers, creditors, consumers, and employees, government) to initiate green activities in firms’ vision to reduce environmental issues is one of the main reason (3). Some of prominent green practices such as Corporate Social Responsibility (CSR) and Green Shared Vision (GSV), green human resource management practices (GHRM) are considered imperative for the progress of the companies and sustainability of the environment and to enhance voluntary green work behavior (VGWB).
Second point: Be clear in the parts 3.2 and 3.3. about the sample. I think its 500 individuals from 100 firms (clearly explain this, 5 from each or random).
Response: thanks for the valuable comment. Yes the information in section 3.2 and 3.3 has been corrected. The sample size is 500 and employees were selected randomly. We have highlighted the corrections in part 3.2 and 3.3 in green color.
Part 3.2:
Total number of firms included were 100, information of the firms included in the study were cement (10), sugar (26), leather (22), steel (8), food (21), beverages (2), furniture (3), and construction (2), pharmaceutical(2), plastic(2), and dairy(2).
Part 3.3:
Total 500 completed questionnaires were received and used in the statistical analysis. Total 700 questionnaires were distributed among the employees and 500 completed questionnaires were used in the analysis thus, yielding the response rate of 71.42%.
The choice of PLS_SEM for this data set is a good choice. The discussion and conclusions are ok!
Response: Thank you so much respected reviewer.
Best Regards,
I think the text might need minor English editing, especially the abstract and introduction's first part.
Response: thank you so much the first paragraph text in abstract and introduction has been corrected.
Response to Suggestions of the reviewer given in Pdf file
What is the main question addressed by the research?
Response: Respective reviewer the following main research question is given at the end of the introduction:
The present study has been conducted to answer the following main research question: Does VGWB can be enhanced by adapting GHRM practices, green shared vision and corporate social responsibility?
- Do you consider the topic original or relevant in the field? Does it
address a specific gap in the field?
As the authors have mentioned at the end of the Introduction part, studies in the field are limited and their perspective is original.
Response: Thank you so much respected reviewer and appreciated
- What does it add to the subject area compared with other published
material?
Yes, it is discussed in the fifth part (Discussion) of the paper however as in every paper this part can always be improved but I think it’s ok! In this form. In the discussion, they’ve mentioned the works with similar and opposite results and their own remarks and opinions in the discussion and conclusions are acceptable.
Response: Thank you so much respected reviewer.
- What specific improvements should the authors consider regarding the
methodology? What further controls should be considered?
I’ve used PLS_SEM many times in my papers and I think the tests done before the SEM and the results are good. It is a good for measuring similar effects with similar (or more) variables. The sample formed from 500 is good.
Response: Thank you so much respected reviewer
- Are the conclusions consistent with the evidence and arguments presented
and do they address the main question posed?
Yes.
Response: Thanks again and appreciated
- Are the references appropriate?
Yes.
Response: Thank you so much and appreciated
- Please include any additional comments on the tables and figures.
The figure 1 explains the model however it can be clearer.
Response: Figure 1 has been improved, now figure 1 is clearer and easy to understand by the readers. Relationships using arrows have been simplified. Font size are unified